# Evaluation of the Safety and Efficacy of Repeated Mesenchymal Stem Cell Transplantations in ALS Patients by Investigating Patients’ Specific Immunological and Biochemical Biomarkers

**DOI:** 10.3390/diseases12050099

**Published:** 2024-05-12

**Authors:** Zahraa Alkhazaali-Ali, Sajad Sahab-Negah, Amir Reza Boroumand, Najmeh Kaffash Farkhad, Mohammad Ali Khodadoust, Jalil Tavakol-Afshari

**Affiliations:** 1Department of Immunology, Immunology Research Center, Faculty of Medicine, Mashhad University of Medical Sciences, Mashhad 9177948959, Iran; zhra18645@gmail.com (Z.A.-A.); kaffashn971@mums.ac.ir (N.K.F.); khodadoustm1@mums.ac.ir (M.A.K.); 2Student Research Committee, Faculty of Medicine, Mashhad University of Medical Sciences, Mashhad 9177948959, Iran; 3Neuroscience Research Center, Mashhad University of Medical Sciences, Mashhad 9177948959, Iran; sahabnegahs@mums.ac.ir (S.S.-N.); arboroumand@ymail.com (A.R.B.); 4Shefa Neuroscience Research Center, Khatam Alanbia Hospital, Tehran 1708310, Iran

**Keywords:** ALS, tumor necrosis factor-alpha, GDNF, creatine kinase, ferritin, BM-MSC

## Abstract

Background: Amyotrophic lateral sclerosis (ALS) is an incurable disease. There are vigorous attempts to develop treatments to reduce the effects of this disease, and among these treatments is the transplantation of stem cells. This study aimed to retrospectively evaluate a mesenchymal stem cell (MSC) therapy cohort as a promising novel treatment modality by estimating some additional new parameters, such as immunological and biochemical factors. Methods: This study was designed as an open-label, one-arm cohort retrospective study to evaluate potential diagnostic biomarkers of repeated infusions of autologous-bone marrow-derived mesenchymal stem cells (BM-MSCs) in 15 confirmed patients with ALS, administered at a dose of 1 × $10^{6}$ cells/kg BW with a one-month interval, in equal amounts in both an intravenous (IV) and intrathecal (IT) capacity simultaneously, via various biochemical (iron (Fe), ferritin, total-iron-binding capacity (TIBC), transferrin, and creatine kinase (CK)) and immunological parameters (tumor necrosis factor-alpha (TNF-α), neurofilament light chain (NFL), and glial-cell-derived neurotrophic factor (GDNF) levels, evaluated during the three-month follow-up period in serum and cerebrospinal fluid (CSF). Results: Our study indicated that, in the case of immunological biomarkers, TNF-α levels in the CSF showed a significant decrease at month three after transplantation compared with levels at month zero, and the *p*-value was *p* < 0.01. No statistically significant changes were observed for other immunological as well as biochemical parameters and a *p*-value of *p* > 0.05. Conclusions: These results can indicate the potential benefit of stem cell transfusion in patients with ALS and suggest some diagnostic biomarkers. Several studies are required to approve these results.

## 1. Introduction

Amyotrophic lateral sclerosis (ALS) is a disease characterized by an adult neurodegenerative onset with progressive deterioration and the loss of cortical and spinal motor neurons. The disease begins as insidious localized muscle weakness and gradually spreads to impair most skeletal muscles, eventually resulting in total paralysis, and death can occur after a few years of diagnosis due to respiratory failure [1]. Frontotemporal dementia (FTD), which affects 20% to 50% of ALS patients, has been strongly linked to the disease in recent years [2]. There are several genes that contribute to developed ALS, such as superoxide dismutase (SOD1), fused in sarcoma (FUS), chromosome 9 open-reading frame 72 (C9Orf72), and transactive response DNA-binding protein 43 (TDP-43), found in different percentages across family and sporadic cases in ALS patients [3,4,5,6]. To date, there has been no defined treatment for this disease, and the available drugs have complementary roles without desirable outcomes [7]. Patients are often misdiagnosed with other conditions before ALS is finally identified [8].

Therefore, it is urgent to find new treatment strategies and specific prognostic as well as diagnostic biomarkers to speed up treatment in this area. Stem cell therapy is a promising approach in this course [9]; stem cell transplantation is extremely promising and might help those with ALS through various mechanisms, such as replacing damaged or lost cells, introducing elements that will have neuroprotective effects, or modifying the pathogenic pathways connected to the toxic microenvironment [10], and, as stated in a previous publication, we clinically established the effectiveness of stem cell transplantation in fifteen patients through the values of ALSFRS and FVC, the ALSFRS score is a validated test, evaluating a patient’s functional status and the disease progression of ALS, which is significantly associated with the quality of life of the patient [11]; the FVC value is another indicator of patient survival and disease progression, indicating respiratory function in affected individuals [12]. These indicative values significantly decreased 6 months after transplantation [13]. Regarding our previous project, and for closer evaluation, in this study, we investigate new additional parameters (immunological and biochemical factors) for our previous samples.

As we know, multiple mechanisms can lead to ALS progression, ranging from the molecular to the cellular level, like RNA and protein metabolism, axonal transport, oxidative stress (OS), and mitochondrial dysfunction [14,15]. Thus, investigating some related immunological and biochemical parameters might help those with ALS.

One of the parameters enrolled in this study is TNF-α, which plays an important role in neuroinflammation. Neuroinflammation is an important feature that is noted in different neurodegenerative diseases, like Parkinson’s disease, multiple sclerosis, and amyotrophic lateral sclerosis [16]. Additionally, neuroinflammation is the inflammation of neurons caused by microglia, which secrete proinflammatory cytokines like TNF-α, interleukin (IL-6), and reactive oxygen [17,18]. In ALS, many cells, like astrocytes and microglia, can express receptors for TNF-α and excrete this marker [19].

Other significant factors that can be exploited to help identify ALS are neurofilaments (NFs), consisting of heavy (H), medium (M), and light (L) heteropolymer-based intermediate filament proteins [20], which are expressed by many neurons and participate in intracellular signaling, structural support, and axon development regulation [21]. The fact that these markers’ levels correspond with disease progression and survival rates, both in the serum and the CSF, has attracted the most attention [22].

A neuroprotective factor that impacts several neurodegenerative diseases and is also enrolled in this study is glial-cell-derived neurotropic factor (GDNF) [23], which increases during embryonic development and gradually decreases in level during adulthood [24]; it can participate in axon regeneration, neuropathic pain reduction, and inflammation reduction [25].

On the other hand, iron metabolism abnormality can arise and participate in the progression of this disease. Iron and its various forms, such as ferritin and total iron binding capacity (TIBC), are other indicators of ALS; many studies have shown that iron can be collected in the spinal cord and cerebrospinal fluid of ALS patients [26], as well as within the skeletal muscles in rat models with ALS [27], which can play a critical role in the central nervous system for various metabolic processes, notably oxidative phosphorylation, myelination, the production of neurotransmitters, and oxygen transport [28]. The ferritin storage form of iron can prevent iron from creating reactive oxygen species (ROS) [29], and in terms of TIBC, which indicates how much transferrin binds with Fe, an elevated level of transferrin occurs when the iron store (ferritin) is depleted, so TIBC increases in patients with an Fe level below the normal range [30].

Many symptoms can further help in the early identification of ALS, such as muscular weakness [31]. Significant changes in serum CK are frequently regarded as a sign that CK-rich tissues, notably muscle, have been injured [32]. In ALS patients, several studies noted an increase in the level of this biomarker due to subsequent muscle breakdown brought on by degenerative changes [33,34].

Concerning the above information, this study focused on estimating the effectiveness of using stem cells as therapy to help address the decline caused by damaged neurons by retrospective cohort estimating some immunological and chemical parameters that can help us to interpret this type of therapy clinically.

## 2. Materials and Methods

### 2.1. Study Design

This retrospective cohort study was single-center and open-label, without a control group, carried out in 15 defined patients with ALS confirmed by a neurologist’s diagnosis and according to El Escorial criteria [35]. All international standards were followed during the execution of the trial at Pastoor Hospital, Iran. All patients were in the age range of 23–60 years, with an FVC > 65% and a history of ALS disease for at least 1 year at the time of admission, and signed an informed written consent form. Additionally, ventilator dependency, any malignancy, or infection 1 week before transplantation were considered exclusion criteria [13]. Based on a previous study [13], in brief, the patients’ bone marrow was aspirated under sterile conditions and local anesthesia; a specialist doctor aspirated the samples, and the patients’ aspiration sites were examined the day after withdrawal to rule out any adverse reactions. After this, stem cells underwent the culturing processing, and the samples were then tested for sterility using endotoxin, reverse transcription–polymerase chain reaction (RT-PCR), and Bactec tests. Stem cells were transplanted to patients simultaneously in intrathecal (IT) (via a lumber puncture in the L3–L4 region, suspended in 2 mL of normal saline) and intravenous (IV) (suspended in 50–60 mL of normal saline) routes at a rate of 1 × $10^{6}$ cells/kg BW [13]. Additionally, for biological evaluation, blood samples from patients were collected three times (months −1 (1 month before the intervention), 1, and 3 months after transplantation), and CSF samples were collected three times (0 (before transplantation), 1, and 3 months after transplantation) [13]. The samples were preserved in the cell bank (−80) until the time of use. At the start of the work day, the samples were brought to the place designated for them for thawing at room temperature, and then work began.

### 2.2. Immunological Factor Estimation

This study was estimated to have three immunological factors: TNF-α, NFL, and GDNF. Estimation was performed using an enzyme-linked immunosorbent assay (ELISA) kit for each factor (ZellBio GmbH, Origen; Germany). The procedures for these three immunological factors were as follows:

For the blank well, only anti-TNF-a antibodies labeled with biotin or streptavidin-HRP were added to the comparison blank well, except for the chromagen solution, and further processing took the same following steps. In the case of the standard well, 50 µL of streptavidin-HRP was added.

For the sample well, a 40 µL sample with 10 µL of TNF-a antibodies, as well as 50 µL of streptavidin-HRP, was added. All wells were covered with a seal plate membrane. The plate was shaken gently and incubated at 37 °C for 60 min; after that, the wells were washed four times in the diluted washing solution for 30 s each time, and 100 µL of chromagen solution was added and then incubated for 10–20 min away from light at 37 °C. Finally, 50 µL of stop solution was added, and the plate was read at 450 nm.

### 2.3. Biochemical Evaluation

Serum iron (Fe), transferrin, ferritin, total iron binding capacity (TIBC), and serum creatine kinase (CK) were evaluated in our study.

The ELISA kit was used for ferritin estimation (DiaMetra, Italy); 10 µL came from the sample, control, and calibrator added into wells, with 200 µL of the conjugate also added to these wells; then, the microplate was incubated at room temperature (22–28 °C) for 1 h. After this, the wells were washed with 300 µL of diluted wash solution five times, and the TMB substrate (100 µL) was added with incubation for 15 min at room temperature in the dark, and finally, the stop solution (100 µL) was added and the microplate was read at 450 nm.

For CK and iron, reagents from Parsazmun, Iran, were used, and TIBC (biorexfars, Iran) was tested on an autoanalyzer (Tokyo Boeki Medisys Inc., Hino, Japan). Iron was read at 600 nanometers (nm), TIBC at 660 nm, and CK at 340 nm. Transferrin saturation as a percentage was calculated as serum iron/TIBC ×100 [36].

### 2.4. Outcomes

In this study, we investigate the safety of the repeated transplantation of BM-MSCs in ALS patients by retrospective cohort evaluating specific immunological and biochemical factors to find diagnostic and/or prognostic biomarkers in this area.

### 2.5. Statistical Analysis

Statistical analysis was performed using the SPSS 23.0 software package (SPSS GmbH Software, Germany). Testing normality was carried out by using the Shapiro–Wilk test. Generalized linear model (GLM) and repeated measures ANOVA (RM ANOVA) analyses were used to test the effect of time on normally distributed data. Non-normally distributed continuous variables are reported as the median (interquartile range, IQR). The comparison of the medians between two related groups was carried out via paired T-tests or Wilcoxon signed-rank tests. Continuous variables are expressed as the mean (for normally distributed data) or the median (for non-normally distributed data) ± SEM (standard error of the mean/median). The comparison of the statistical differences in the patients’ data was performed 3 times (before the intervention, and at months 1 and 3) for serum, and at months 0, 1, and 3 for CSF samples. *p*-values less than 0.05 were considered to indicate significance. The statistical graphs were created using the Graph Pad Prism8.1 software program (Graph Pad Software, San Diego, CA, USA).

## 3. Results

### 3.1. Patients’ Characterization and Safety Evaluation

For a safety evaluation of patients’ demographic information, please refer to our previous study [13].

### 3.2. Immunological Assessment

Changes in the levels of specific cytokines were investigated three times (months −1, 1, and 3) in serum samples and three times in CSF samples (months 0, 1, and 3) from patients, and the results are as follows.

#### 3.2.1. Tumor Necrosis Factor-Alpha (TNF-α) Levels

As presented in Figure 1A, the initial measurement showed that the serum levels of the TNF-α cytokine in patients one month before the intervention (month −1) were 16.19 ± 1.05, compared with one month (16.95 ± 0.40) and three months (15.44 ± 1.65) after cell transplantation; the *p*-values were 0.68 and 0.66, respectively, which did not change significantly and remained constant.

Furthermore, as shown in Figure 1B, the amount of this cytokine in the CSF of the patients remained statistically constant at month 1 (20.29 ± 0.68) compared with the beginning of the experiment (19.46 ± 0.41); however, significant differences were observed between month 3 (16.41 ± 1.49, *p* = 0.003) and month 0 (baseline), as well as between month 1 and month 3 (*p* = 0.007), respectively.

#### 3.2.2. Neurofilament Light Chain (NFL) Levels

The analysis of NFL serum levels showed statistically nonsignificant differences (*p* = 0.90) at the end of the study (7.62 ± 1.00) compared with the baseline (7.75 ± 0.82), and this was the same when comparing month −1 with month 1 (8.49 ± 0.25), (*p* = 0.56). Indeed, as shown in Figure 1C, the serum levels of this cytokine remained stable during the three-month follow-up period of patients after cell transplantation.

Furthermore, the measurement of this cytokine in the patients’ CSF during the follow-up period showed a nonsignificant decrease (*p* = 0.23) in its levels at the end of the study (8.28 ± 0.27) compared with the baseline (9.41 ± 0.38). The same results noted in the case of month one (8.31 ± 0.94) compared with month three were observed (Figure 1D).

#### 3.2.3. Glial-Cell-Derived Neurotrophic Factor (GDNF) Levels

The serum measurement of GDNF cytokine levels showed that there was no statistically significant change (*p* = 0.90, 0.56) in the amount of this cytokine at the beginning (2.43 ± 0.17) and the end of the experiment (2.44 ± 0.25) and after one month of receiving the MSCs (2.47 ± 0.23) (Figure 1E).

Also, as shown in Figure 1F, a slight and nonsignificant increase (*p* = 0.21) in month three of transplantation in the amount of this cytokine (3.20 ± 0.51) compared with the initial amount (2.90 ± 0.02) was observed in the level of the CSF during the follow-up period. For month one (2.91 ± 0.18), nonstatistical differences were observed compared with the baseline.

### 3.3. Biochemical Evaluation

Changes in the level of specific biochemical factors were investigated three times (months −1, 1, and 3) in serum samples from the patients, and the results are as follows.

#### 3.3.1. Iron (Fe) Levels

The statistical analysis of iron (Fe) levels with a normal range of 23–149 µg/dL showed that the level of this factor in month −1 (74.55 ± 9.42) in comparison with month 1 (67.40 ± 9.16) and month 3 (56.85 ± 9.91) did not change statistically; *p* = 0.62 and 0.19, respectively. However, this test showed a gradual decrease in means over these three months, but it was still within the normal range, as mentioned above (Figure 2A).

#### 3.3.2. Ferritin Levels

The statistical results of ferritin with the normal range of 16.8–276.4 ng/mL show that this factor had a decreasing trend during the three months, such that its amount at the baseline (383.62 ± 92.73) had a statistically nonsignificant difference from that at month 3 (266.27 ± 67.07, *p* = 0.34), and this was the same for month 1 (375.40 ± 109.25, *p* = 0.95) (Figure 2B).

#### 3.3.3. Total Iron Binding Capacity (TIBC) Levels

Analysis of TIBC serum levels with the normal range of (239–450 µg/dL) showed statistically nonsignificant differences (*p* = 0.61) at the end of the study (296.60 ± 14.01) compared with the baseline (277.26 ± 25.15). A nonsignificant increase was also observed during month one (335.36 ± 109.25) (Figure 2C).

#### 3.3.4. Transferrin Levels

For transferrin in the normal range of 25%–35% [37], a gradually nonsignificant decrease in mean percentage was observed compared with month −1 (27.29 ± 3.34) for month 1 (20.42 ± 3.26) and month 3 (19.96 ± 3.81). The *p*-values were 0.21 and 0.14, respectively (Figure 2D).

#### 3.3.5. Creatine Kinase (CK) Levels

As shown in Figure 2E, there was a nonsignificant difference in serum CK levels (24–195 µ/L) in patients before an injection (391.77 ± 152.59) and after one (383.80 ± 195.35, *p* = 0.97) and three (270.13 ± 171.71, *p* = 0.60) months of stem cell transplantation, respectively.

## 4. Discussion

As we know, there are no available treatments for ALS disease; in the presented study, we retrospectively evaluated the effectiveness of stem cell transplantation in ALS cases as therapy in a cohort through the estimation of the concentrations of some specific immunological and biochemical factors in serum and CSF.

Several studies have shown the importance of stem cell transplantation. MSCs can play a vital role in controlling the immune response through the upregulation of neuroprotective cytokines like IL-10 and downregulating proinflammatory cytokines, TNF-α and IL-1, by releasing neurotropic factors and enhancing neurogenesis [38], and this is the same as for different neurological diseases [39,40]. Others showed that endogenous neural stem cells (NSCs) can differentiate directly into a mature neuron or indirectly into neural progenitor cells (NPCs) [41]. According to a study by Zhang et al. in 2016, using NSC implantation led to a dramatic decline in microgliosis and glial markers, proinflammatory cytokine (TNF-α, IL-6, and IL-1) expression, and toll-like receptor 4 (TLR4) [42].

Our investigation showed the outcomes regarding TNF-α, which decreased in concentration in the case of the CSF and was stable in serum samples. One of the important features of muscle atrophy is the destruction of the neuromuscular junction, which leads to motor neuron depletion and the accumulation of macrophages, as well as other immunological elements, such as TNF-α, which plays a principal role in the relapsing of motor neurons [43].

For NFL, this study indicated the stability of NFL levels after transplantation through months in serum and the CSF. One study showed that when MSC-NTF cells were administered, it led to a decrease in the levels of NFL and pNfH in ALS participants throughout the experiment [44]; another revealed that the level of NFL would not change in the early trial of studied ALS patients [45]. However, when ALS progression was noted, there were increased levels of the NFL in the blood and CSF [46], highlighting the severe impairment of motor neurons and axons [47], and low levels of the NFL were linked to functional decline being slower [48]. Other researchers showed that the NFL and phosphorylated NFL (p-NFL) were estimated in CSF samples in other neurological diseases; for example, patients treated with stem cells for MS revealed stability in concentration [49]. In contrast, others showed that the estimated level of CSF NFL was reduced [50].

As another immunological factor, we evaluated GDNF levels in our study after MSC transplantation, which showed a nonsignificantly increasing trend in serum and CSF samples through the months, which is another promising result. The administration of hematopoietic mesenchymal stem cells (hMSCs) intraspinally in an ALS rat model increased the rate of survival and improved motor neuron (MN) viability, possibly via upregulating glial-cell-line-derived NTF (GDNF) [51,52]. Axon regeneration, a reduction in neuropathic pain, a reduction in cortical infarction, neurogenesis, and a reduction in inflammation are all facilitated by GDNF [25].

In addition, this study showed a decreasing trend in iron (Fe), but it remained within the normal range. It is well recognized that changes in cellular iron metabolism can cause oxidative stress (one of the main causes of ALS pathogenesis) [53]. Iron can be collected in the spinal cord and cerebrospinal fluid in ALS patients [26]; some studies showed an increased level of Fe in the plasma [54] and CSF [55] of ALS patients. The decreasing trend in Fe levels after MSC transplantation in our study may be due to the antioxidant properties of these cells [56], or it may be due to patients with ALS having difficulty with the swallowing process, and this leads to a rapidly deteriorating nutrition status due to insufficient iron uptake [57].

Increased levels of ferritin may be due to inflammation that occurs through ALS. It is possible to consider ferritin as an indicated marker of inflammation related to the severity of the disease [58]. In this study, there was a decrease in the level of ferritin throughout the experiment, despite these levels being above the normal range and TIBC still being within the normal range. According to Goodall et al., an increased serum ferritin level may indicate elevated stored iron in the body or be a consequence of muscle deterioration [59]. A meta-analysis study found the possibility of increased levels of ferritin, CK, the transferrin saturation coefficient (TSC), and fasting blood glucose (FBG), as well as a decrease in the level of TIBC, in ALS cases in comparison with a control [53]; this may indicate that ALS diseases have multifactorial causes with degeneration of motor neuron activation. Furthermore, our study indicated a nonsignificant decrease in the transferrin saturation level after three months of transplantation. The iron homeostasis disruption could result from the activation of the immune system, which is considered one of ALS’s pathogeneses; biochemical tests can definitely help in the early estimation of the changes that happen within these patients with ALS and may be good parameters in interpreting the early signs of malnutrition that occur during the disease’s development [60].

Additionally, changes in muscle mass are one of the signs of ALS, and they are one of the biochemical parameters that can be assessed in this case is CK. According to the study by Hertel et al., an increased CK level can increase survival time and provide a good interpretation of the ALSFRS-R score [61]; however, a decreased level was indicated in the case of muscle atrophy [34].

Our result shows a gradual decrease in CK (but still above the normal range) after MSC transplantation, and ALS-FRS and FVC showed stability after three months of injection.

This study has some limitations, such as a small sample size, a need for a control group, and a short follow-up period. Larger clinical trials with a control placebo group can confirm these results with precision.

## 5. Conclusions

According to this retrospective cohort study, there was a significant decrease in the level of TNF-α in the CSF of ALS patients after repeated MSC transplantation after a three-month follow-up period. Additionally, the concentrations of the NFL and GDNF showed stability in their levels. Specific biochemical markers also changed favorably but not significantly. To sum up, overall improvement in patient status after cell therapy might be a promising result of this study. More studies are needed, including those with a greater number of participants and a longer follow-up period, to adequately substantiate the advantageous long-term effectiveness of our strategy in slowing disease development.

## Figures and Tables

**Figure 1 diseases-12-00099-f001:**
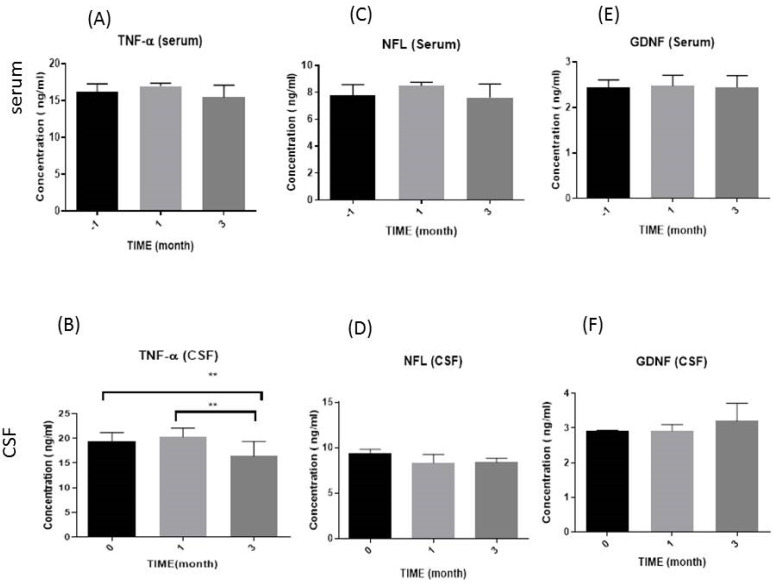
Schematic diagram of the immunological parameters through months in serum and the CSF (**A**,**B**): tumor necrosis factor-alpha (TNF-α); (**C**,**D**): neurofilament light chain (NFL); and (**E**,**F**): glial-cell-derived neurotropic factor (GDNF), respectively. In serum, tested three times (before, as month −1, and after 1 and 3 months of transplantation). In CSF, tested before (month 0), after 1, and 3 months after transplantation. Levels tested in ELISA kit, ** (*p*-value < 0.01). ng/mL; nanograms per milliliter. CSF; cerebrospinal fluid (CSF).

**Figure 2 diseases-12-00099-f002:**
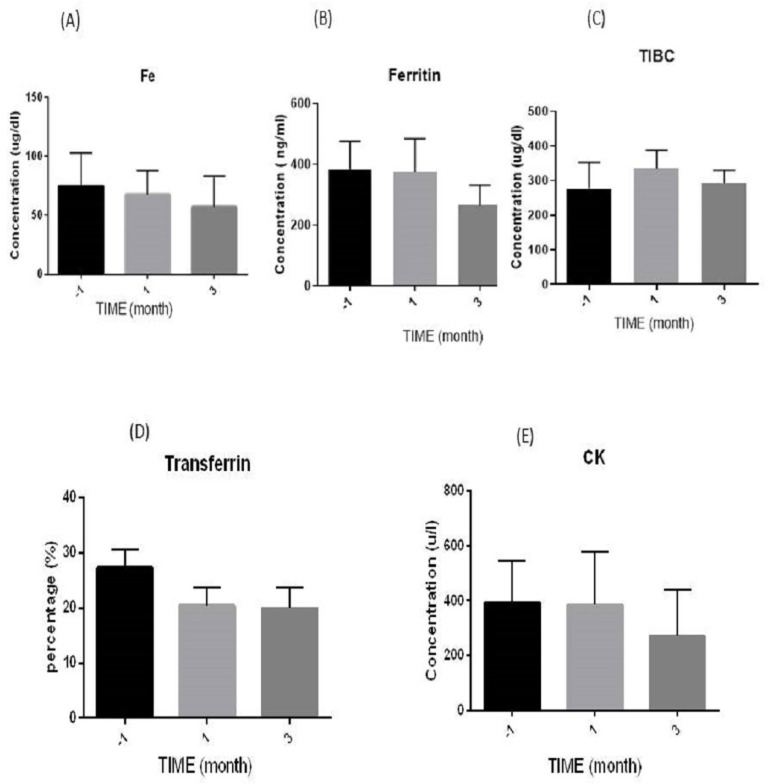
Schematic diagram of the biochemical serum levels during the study: (**A**) iron (Fe), (**B**) ferritin, (**C**) total iron binding capacity (TIBC), (**D**) transferrin (as percentage, %), and (**E**) creatine kinase (CK). The three times estimated were before (month −1) and after one month, as well as three months after transplantation. *p*-value > 0.05. µg/dL: microgram per deciliter; u/L: units of enzyme activity per liter; and ng/mL: nanograms per milliliter.

## Data Availability

The data used in this study can be requested from the corresponding author.

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
