# Peer review of "Evaluation of the Safety and Efficacy of Repeated Mesenchymal Stem Cell Transplantations in ALS Patients by Investigating Patients’ Specific Immunological and Biochemical Biomarkers"

_diseases, 2024, doi:10.3390/diseases12050099_

Round 1
Reviewer 1 Report
Comments and Suggestions for Authors
See the file please.

Already commented in the file.
Author Response
Response to Reviewer 1 Comments
|
||
Summary |
|
|
Thank you very much for taking the time to review this manuscript. Please find the detailed responses below and the corresponding corrections highlighted in the re-submitted files. |
||
|
|
|
|
|
|
|
|
|
|
|
|
|
|
|
|
|
|
|
|
|
A point-by-point response to Comments |
||
Comments 1: About the correlation analysis, which data set was utilized for correlation analysis, 1 month or 3 months?
|
||
Response 1: [Thank you for pointing this out. The correlation has been made in serum between biochemical parameters and immunological parameters in general terms throughout the month overall, and we did not specify a specific month.
Comments 2: Some paragraph breaks are not needed. For example: the break at Line 301-302. Response 2: [Thank you for pointing this out. we fixed it throughout the article ]
|
||
Comments 3: Result section line 165, shouldn’t it be “(P-value<0.05) as published previously |
||
Response 3: Thank you for pointing this out, we have rephrased this part in our manuscript as 3. Results 3.1. Patients' Characterization and Safety Evaluation (For a safety evaluation of patients' demographic information, please refer to our previous study [13].)
|
||
Response to Comments on the Quality of English Language |
||
Point 1: The writing of this manuscript needs to be edited by an English reader. Numerous errors in grammars and typos. Some expressions are even not right. For example, • In Discussion section line 316, ….an increased serum ferritin level may indicate elevated stored iron in the body or result in muscle deterioration. I assume it should be “or a consequence of muscle deterioration.” • Or in the Abstract, “Numerous approaches have been made to meet this impact.” What does “impact” mean in this case? • Or like the lines 48-50, “Taking into account…..immune dysfunction.” It is even not a sentence but a group of phrases.
|
||
Response1: Thank you for pointing this out . We fixed it in our article (an increased serum ferritin level may indicate elevated stored iron in the body or be a consequence of muscle deterioration), also the article was edited for Language by (Author Services Language Editing) in MDPI.
Point 2: Or in the Abstract, “Numerous approaches have been made to meet this impact.” What does “impact” mean in this case? Response 2: (Thank you for pointing this out, and we have revised this sentence in our article,( Background: Amyotrophic lateral sclerosis (ALS) is an incurable disease. There are vigorous attempts to develop treatments to reduce the effects of this disease, and among these treatments is the transplantation of stem cells.) and article edited for English language by (Author Services Language Editing) in MDPI point 3 : or like the lines 48-50, “Taking into account…..immune dysfunction.” It is even not a sentence but a group of phrases. Respons 3: Thank you for pointing this out, and we have revised this sentence in our article, and article edited for English language by (Author Services Language Editing) in MDPI
|
||
Additional clarifications |
||
We would like to clarify to you that a request has been made from us by the Diseases Editorial Office (Increase the number of words in the article) and we have lengthened the article by adding sentences and paragraphs throughout the article (introduction, material and method (procedure of ELISA test for three immunological parameters, and for ferritin test ), discussion) that are appropriate to the topic, also we rephrase the material and method part and make it in one section, also according to a request has been made from us by the Diseases Editorial Office.
Thank you for your consideration
|

Reviewer 2 Report
Comments and Suggestions for Authors
This manuscript is a follow-up study of a published study on stem cell transplantation in ALS (Tavakol-Afshari et al, 2021, Regen Ther, doi: 10.1016/j.reth.2021.07.006). The authors use blood and CSF samples which have been obtained during the published study for the evaluation of additional parameters, in particular the cytokines TNF-α and GDNF, the concentration of Neurofilament light chain NF-L, Serum iron parameters and creatinine kinase CK. They show that all of these parameters remain at a stable expression level during the 3 months observation period, and TNF-α shows a small reduction at the 3 months timepoint.
The manuscript provides a small but interesting addition to the published data of the study, confirming that there were no adverse effects and providing some evidence that the disease progression was slow in the 3 months observation period after the stem cell transplantation.
Unfortunately, the presentation of the results and the organization of the manuscript are somewhat unfortunate and should be modified.
1) The manuscript presents the findings as if they were the results of an independent study. If this was the case, the documentation of the study would be insufficient. There is, e.g. no reference to the approval of the study by an ethical committee etc.
It should, therefore, be clearly spelled out that all measurements were done on material and samples which have already been secured in the published study. In this paper, there is also a satisfactory discussion of the ethical questions related to this study.
2) Accordingly, the title, abstract should be changed making this point clear. The introduction should be changed an should refer more clearly to the published study, clearly spelling out that now additional parameters in blood and CSF sample from the previous study were measured.
3) The methods should refer to the published study for the study design etc. and should focus mostly on the measurements (current points 2.4 -2.8). Points 2.1 to 2.4 should be removed and replaced by one paragraph referring to the published study.
4) The same applies to point 3.1, which belongs to the published study.
5) Point 3.4 should be removed. These correlations have no biological relevance.
6) The discussion should be shortened and focus on the studied serum and CSF parameters.
Minor points:
line 53 and 54: Furthermore, one of the primary proinflammatory cytokines produced by reactive glia and leading to neuroinflammation are …
lines 164 and 165: P-value ><0.05
Comments on the Quality of English LanguageSee minor points for examples of language problems. Similar problems are present throughout the manuscript.
Author Response
Response to Reviewer 2 Comments
|
||
Summary |
|
|
Thank you very much for taking the time to review this manuscript. Please find the detailed responses below and the corresponding revisions changes in the re-submitted files. |
||
|
|
|
|
|
|
|
|
|
|
|
|
|
|
|
|
|
|
|
|
|
Point-by-point response to Comments |
||
Comments 1: The manuscript presents the findings as if they were the results of an independent study. If this was the case, the documentation of the study would be insufficient. There is, e.g. no reference to the approval of the study by an ethical committee etc.
It should, therefore, be clearly spelled out that all measurements were done on material and samples which have already been secured in the published study. In this paper, there is also a satisfactory discussion of the ethical questions related to this study.
|
||
Response 1: [Thank you for pointing this out. Our study as retrospective study as written in the name of the article(Evaluation of Safety and Efficacy of Repeated Mesenchymal Stem Cell Transplantations in ALS Patients by Investigating Patient's Specific Immunological and Biochemical Biomarkers: A Retrospective study), and when we mentioned it for first time submission , in section (the material and methods) we have referred to the previous study[17] as reference number 17 ( in the submitted file for the first time ) Also mentioned the approval of the study by an ethical committee in the section (Institutional Review Board Statement) : This study was achieved by the rules and morals of the Helsinki Medical Principle of Clinical Practice and under the direction of the Ethics Committee of the Mashhad University of Medical Sciences, Mashhad, Iran. (Reg. No. IR.MUMS.REC.1399.269). Furthermore, this clinical trial was registered with the Iranian Organization for Clinical Trials (ID: IRCT20160809029275N2). Also section material and method was rephrased and now we have referred to the previous study as reference [13].
|
||
Comments 2: Accordingly, the title, abstract should be changed making this point clear. The introduction should be changed an should refer more clearly to the published study, clearly spelling out that now additional parameters in blood and CSF sample from the previous study were measured.]
|
||
Response 2: Thank you for pointing this out. We believe that the title is currently consistent with what was presented in the article For abstract ; some change was made (Background: Amyotrophic lateral sclerosis (ALS) is an incurable disease. There are vigorous attempts to develop treatments to reduce the effects of this disease, and among these treatments is the transplantation of stem cells. This study aimed to retrospectively evaluate mesenchymal stem cell (MSC) therapy as a promising novel treatment modality by estimating some additional new parameters, such as immunological and biochemical factors…..) And this is the same for introduction (and, as stated in a previous publication , we clinically established the effectiveness of stem cell transplantation in fifteen patients through the values of ALSFRS and FVC, the ALSFRS score is a validated test, evaluating a patient's functional status and the disease progression of ALS, which is significantly associated with the quality of life of the patient [11]; the FVC value is another indicator of patient survival and disease progression, indicating respiratory function in affected individuals [12], these indicative values significantly decreased 6 months after transplantation[13]. Regarding our previous project, and for closer evaluation, in this study we investigate new additional parameters (immunological and biochemical factors) for our previous samples.) .please refers to re-submitted article for further revision.
Comments 3: the methods should refer to the published study for the study design etc. and should focus mostly on the measurements (current points 2.4 -2.8). Points 2.1 to 2.4 should be removed and replaced by one paragraph referring to the published study. Response 3: Thank you for pointing this out. Points 2.1 to 2.4 we replaced in one section and we also refer to published study. (2. Materials and Methods 2.1. Study Design This retrospective trial study was single-center and open-label, without a control group, carried out in 15 defined patients with ALS confirmed by a neurologist's diagnosis and according to El Escorial criteria [35]. All international standards were followed during the execution of the trial at Pastoor Hospital, Iran. All patients were in the age range of 23–60 years, with an FVC > 65% and a history of ALS disease for at least 1 year at the time of admission, and signed an informed written consent form. Additionally, ventilator dependency, any malignancy, or infection 1 week before transplantation were considered exclusion criteria [13]. Based on a previous study [13], in brief, the patients’ bone marrow was aspirated under sterile conditions and local anesthesia; a specialist doctor aspirated the samples, and the patients' aspiration sites were examined the day after withdrawal to rule out any adverse reactions. After this, stem cells underwent the culturing processing and the samples were then tested for sterility using endotoxin, reverse transcription-polymerase chain reaction (RT-PCR), and Bactec tests. Stem cells were transplanted to patients simultaneously in intrathecal (IT) (via a lumber puncture in the L3–L4 region, suspended in 2 ml of normal saline) and intravenous (IV) (suspended in 50–60 ml of normal saline) routes at a rate of 1× ã€–10〗^6cells/kg BW [13]. Additionally, for biological evaluation, blood samples from patients were collected three times (months -1 (1 month before the intervention), 1, and 3 months after transplantation), and CSF samples were collected three times (months 0 (before transplantation), 1, and 3 months after transplantation). The samples were preserved in the cell bank (-80) until the time of use. At the start of the work day, the samples were brought to the place designated for them for thawing at room temperature, and then work began.)
Comments 4: The same applies to point 3.1, which belongs to the published study Response 4: Thank you for pointing this out. Points 3.1 was removed and this modified in the revised manuscript . (3. Results 3.1. Patients' Characterization and Safety Evaluation For a safety evaluation of patients' demographic information, please refer to our previous study [13].) . Comment 5: Point 3.4 should be removed. These correlations have no biological relevance. Response 5: Thank you for pointing this out. We believe these correlation can give good point for parameters that enrolled in the correlation as positive correlation or as negative correlation , and we mentioned how these parameter possibly work in section discussion . (A positive correlation was noted in this study, interpreted between Fe, CK, and NFL; this may be due to the fact that Fe can play a potential role in neuroinflammation and oxida-tive stress by producing hydroxyl radicals known as reactive oxygen species (ROS) and TNF-α excretion, which lead to the damage and death of neuron cells [62,63]; an increased level of CK may be due to the denervation of muscle breakdown or metabolic changes that occur as part of a protective mechanism or many other active metabolisms in ALS [64], and, as mentioned above, the NFL level increased in cases of ALS onset and can play a significant damaging role in motor neurons……). Please refer to resubmitted article for further discussion. Comments 6: the discussion should be shortened and focus on the studied serum and CSF parameters. Responde 6: Thank you for pointing this out. We also believe this, but due to the multiple prameters that enrolled in this study , It deserves a little bit of length.
|
||
4. Response to Comments on the Quality of English Language |
||
Point 1: minor points for examples of language problems. Similar problems are present throughout the manuscript.
|
||
Response 1: Thank you for pointing this out,and we fixed it throughout the study , the article was editing for Language by (Author Services Language Editing) in MDPI. |
Additional clarifications
We would like to clarify to you that a request has been made from us by the Diseases Editorial Office (Increase the number of words in the article) and we have lengthened the article by adding sentences and paragraphs throughout the article (introduction, material and method (procedure of ELIZA for three immunological prameters and for ferritin ELIZA test ), discussion) that are appropriate to the topic, also we rephrase the material and method part and make it in one section, also according to a request has been made from us by the Diseases Editorial Office.
Thank you for your consideration
Reviewer 3 Report
Comments and Suggestions for Authors
The authors present an interesting small study examining immunological and bichemical biomarkers after stem cell transplantation in ALS. The biggest limitation is sample size (15 patients).
The authors perform a standard statistical analysis that assumes normality to examine changes in biomarkers across 3 time points. However, the small sample size of 15 patients is burying any potential signal in the data. While there is an importance in examining the aggregate trend using paired t-test, it is difficult to interpret the result. It would be helpful if the authors showed their power analysis using the experimental variance to estimate how many patients would be needed to see a significant difference at alpha of 0.05. This extra context would provide more clarity to the results. Another alternative would be to do a Fisher's exact test (better for small sample sizes), but the power analysis would be the easiest to interpret and require a minimum number of changes to the current presentation.
If the authors have access to functional disease measurements that correlate with these time points (ALFRS-R, etc.) that would really improve the potential implications of their work. It appears based on their prior cited work, this data does exist. Mapping the functional course (i.e. potential slower progression) with the biomarkers would greatly strengthen the work.
The authors need to pull back on some of the conclusions wording about the stem cell transplants potential benefit. If the power analysis is added, then maybe the wording can stay as is. But right now the strength of the statements or promise of benefit is stronger than the shown data.
Author Response
Response to Reviewer 3 Comments
|
||
1. Summary |
|
|
Thank you very much for taking the time to review this manuscript. Please find the detailed responses below and the corresponding revisions changes in the re-submitted file.
|
||
|
|
|
|
|
|
|
|
|
|
|
|
|
|
|
|
|
|
|
|
|
2. Point-by-point response to Comments |
||
Comments 1: The authors perform a standard statistical analysis that assumes normality to examine changes in biomarkers across 3 time points. However, the small sample size of 15 patients is burying any potential signal in the data. While there is an importance in examining the aggregate trend using paired t-test, it is difficult to interpret the result. It would be helpful if the authors showed their power analysis using the experimental variance to estimate how many patients would be needed to see a significant difference at alpha of 0.05. This extra context would provide more clarity to the results. Another alternative would be to do a Fisher's exact test (better for small sample sizes), but the power analysis would be the easiest to interpret and require a minimum number of changes to the current presentation.
|
||
Response 1: Thank you for pointing this out. We believe that the type of statistics that was used is consistent with what was presented in the manuscript, also we appreciate this valuable point mentioned and will certainly benefit from it in future research.
|
||
Comments 2: If the authors have access to functional disease measurements that correlate with these time points (ALFRS-R, etc.) that would really improve the potential implications of their work. It appears based on their prior cited work, this data does exist. Mapping the functional course (i.e. potential slower progression) with the biomarkers would greatly strengthen the work. |
||
Response 2: Thank you for pointing this out. Regarding the rules we cannot repeat the data in this study.
Comments 3: The authors need to pull back on some of the conclusions wording about the stem cell transplants potential benefit. If the power analysis is added, then maybe the wording can stay as is. But right now the strength of the statements or promise of benefit is stronger than the shown data. Response 3: Thank you for pointing this out, as you noted may be some results of this study were not significant , but regarding to our previous significant results for ALS-FRS and FVC, and also TNF-α results of this study, in total it seems that this intervention was beneficial. May be evaluating some other parameters in future studies by long term period evaluation can show more positive results in this area, but to until now we writing in the revised manuscript as in abstract; (Conclusions: These results can indicate the potential benefit of stem cell transfusion in patients with ALS and suggest some diagnostic biomarkers. Several studies are required to approve these results) , and in the conclusion section as; (According to this study, there was a significant decrease in the level of TNF-α in the CSF of ALS patients after repeated MSC transplantation after a three-month follow-up period. Additionally, the concentrations of the NFL and GDNF showed stability in their levels. Specific biochemical markers also changed favorably but not significantly. To sum up, overall improvement in patient status after cell therapy might be a promising result of this study. More studies are needed, including those with a greater number of participants and a longer follow-up period, to adequately substantiate the advantageous long-term effectiveness of our strategy in slowing disease development)
|
||
Additional clarifications We would like to clarify to you that a request has been made from us by the Diseases Editorial Office (Increase the number of words in the article) and we have lengthened the article by adding sentences and paragraphs throughout the article (introduction, material and method (procedure of ELISA test for three immunological parameters, and for ferritin test ), discussion) that are appropriate to the topic, also we rephrase the material and method part and make it in one section , also according to a request has been made from us by the Diseases Editorial Office.
Thank you for your consideration. |
Round 2
Reviewer 2 Report
Comments and Suggestions for Authors
The authors now provide a revised version of their manuscript and a rebuttal to the reviewer's comments. Unfortunately, the authors seem to not accept most of the criticism of this reviewer.
They argue, that the term "retrospective study" would imply, that they do additional measurements on samples from patients who were part of the original study. Unfortunately, this is not the meaning of "retrospective study". I copy here the definition of a retrospective study given by NIH (https://www.cancer.gov/publications/dictionaries/cancer-terms/def/retrospective-study):
retrospective study: A study that compares two groups of people: those with the disease or condition under study (cases) and a very similar group of people who do not have the disease or condition (controls). Researchers study the medical and lifestyle histories of the people in each group to learn what factors may be associated with the disease or condition. For example, one group may have been exposed to a particular substance that the other was not. Also called case-control study.
This is not, what the authors do. They use samples data from a previously published interventional study (stem cell transplantation) and do additional measurements. The term of a "retrospective study" in this context is misleading and the authors are advised to get informed about the basic principles of study design.
Previous comment 5: Point 3.4 should be removed. These correlations have no biological relevance
Again, the authors have not responded to the criticism of this reviewer and keep the meaningless correlations between measured serum parameters in the manuscript.
Previous Comment 6: the discussion should be shortened and focus on the studied serum and CSF parameters.
Again, the authors blatantly reject the point of the reviewer with the argument "due to the multiple prameters that enrolled in this study, It deserves a little bit of length".
If authors are not willing or unable to respond to the well-founded criticism of a reviewer, the only consequence can be to recommend rejection of the manuscript.
Comments on the Quality of English LanguageN/A
Author Response
Thank you very much for taking the time to review this manuscript. Please find the detailed responses below and the corresponding in the re-submitted file.
Point-by-point response to Comments |
Comments 1: The authors now provide a revised version of their manuscript and a rebuttal to the reviewer's comments. Unfortunately, the authors seem to not accept most of the criticism of this reviewer. They argue, that the term "retrospective study" would imply, that they do additional measurements on samples from patients who were part of the original study. Unfortunately, this is not the meaning of "retrospective study". I copy here the definition of a retrospective study given by NIH (https://www.cancer.gov/publications/dictionaries/cancer-terms/def/retrospective-study): retrospective study: A study that compares two groups of people: those with the disease or condition under study (cases) and a very similar group of people who do not have the disease or condition (controls). Researchers study the medical and lifestyle histories of the people in each group to learn what factors may be associated with the disease or condition. For example, one group may have been exposed to a particular substance that the other was not. Also called case-control study. This is not, what the authors do. They use samples data from a previously published interventional study (stem cell transplantation) and do additional measurements. The term of a "retrospective study" in this context is misleading and the authors are advised to get informed about the basic principles of study design Response 1: Thank you for pointing this out. We agree with this comment, and we apologize for the confusion regarding this issue, we would like to point out to you that the article was conducted on the basis of (retrospective cohort study),we also fixed that throughout the article. Also we deleted (A rterospective study) from the title of the article, so that we can clear up this confusion that has occurred. Please refer to the revised submitted article Comments 2: Previous comment 5: Point 3.4 should be removed. These correlations have no biological relevance Again, the authors have not responded to the criticism of this reviewer and keep the meaningless correlations between measured serum parameters in the manuscript. Response 2: Thank you for pointing this out. We have removed it from our article, and the discussion paragraph allocated to it has been removed . Please refer to the revised submitted article
Comment 3: Previous Comment 6: the discussion should be shortened and focus on the studied serum and CSF parameters. Again, the authors blatantly reject the point of the reviewer with the argument "due to the multiple prameters that enrolled in this study, It deserves a little bit of length". Response 3: Thank you for pointing this out. We shortened the discussion and it is now only focused on studied serum and CSF parameters .Please refer to the revised submitted article. If authors are not willing or unable to respond to the well-founded criticism of a reviewer, the only consequence can be to recommend rejection of the manuscript. The changes requested by you have been made, and we look forward to your positive response Thank you for your consideration
|
Round 3
Reviewer 2 Report
Comments and Suggestions for Authors
The authors have now tried to address the criticism of the reviewer and have made changes in the manuscript which address the most severe problems. The manuscript stuill suffers from poor writing, the use of inappropriate terms (it is also not a retrospective cohort study) and poor introduction and discussion. However, in its present form it is at least clear, what it is about and it refers appropriately to the previous study from which the serum samples derive. As it adds some additional measurements to to the original study, it is ok to be published as juidged by this reviewer.
Comments on the Quality of English LanguageThe manuscript still needs extensive language editing.